**www.cambridge.org/qrd**

# Energy landscapes and heat capacity signatures for peptides correlate with phase separation propensity

Nicy[1] 🄳, Rosana Collepardo-Guevara[1,2,3] 🄳, Jerelle A. Joseph[4] 🄳 and David J. Wales[1] 🄳

[1]Yusuf Hamied Department of Chemistry, University of Cambridge, Cambridge, UK; [2]Department of Physics, University of Cambridge, Cambridge, UK; [3]Department of Genetics, University of Cambridge, Cambridge, UK and [4]Department of Chemical and Biological Engineering, Princeton University, Princeton, NJ, USA

## Research Article

**Keywords:**
energy landscapes; liquid-liquid phase separation; global optimisation; discrete path sampling

**Corresponding authors:**
Jerelle A. Joseph and David J. Wales;
Emails: jerellejoseph@princeton.edu;
dw34@cam.ac.uk

### Abstract

Phase separation plays an important role in the formation of membraneless compartments within the cell and intrinsically disordered proteins with low-complexity sequences can drive this compartmentalisation. Various intermolecular forces, such as aromatic–aromatic and cation–aromatic interactions, promote phase separation. However, little is known about how the ability of proteins to phase separate under physiological conditions is encoded in their energy landscapes and this is the focus of the present investigation. Our results provide a first glimpse into how the energy landscapes of minimal peptides that contain $\pi$–$\pi$ and cation–$\pi$ interactions differ from the peptides that lack amino acids with such interactions. The peaks in the heat capacity ($C_V$) as a function of temperature report on alternative low-lying conformations that differ significantly in terms of their enthalpic and entropic contributions. The $C_V$ analysis and subsequent quantification of frustration of the energy landscape suggest that the interactions that promote phase separation lead to features (peaks or inflection points) at low temperatures in $C_V$. More features may occur for peptides containing residues with better phase separation propensity and the energy landscape is more frustrated for such peptides. Overall, this work links the features in the underlying single-molecule potential energy landscapes to their collective phase separation behaviour and identifies quantities ($C_V$ and frustration metric) that can be utilised in soft material design.

## Introduction

Biomolecular condensates are membraneless organelles within the cell that are thought to form via phase separation of proteins and nucleic acids (Brangwynne *et al.*, 2009; Banani *et al.*, 2017; Boeynaems *et al.*, 2019; Mittag and Pappu, 2022). Intrinsically disordered proteins are found ubiquitously in naturally occurring phase-separating proteins and the flexible nature of these proteins promotes transient interactions required for phase separation (Jonas and Izaurralde, 2013; Malinovska *et al.*, 2013; Quiroz and Chilkoti, 2015; Schmidt and Görlich, 2015; Uversky *et al.*, 2015; Pak *et al.*, 2016; Harmon *et al.*, 2017; Dignon *et al.*, 2018; Schuster *et al.*, 2020). Mutational studies have shown that $\pi$–$\pi$ (aromatic–aromatic) and cation–$\pi$ (cation–aromatic) interactions promote biomolecular phase separation, especially those involving tyrosine (Y), phenylalanine (F), and arginine (R) (Nott *et al.*, 2015; Brady *et al.*, 2017; Lin *et al.*, 2017; Qamar *et al.*, 2018; Wang *et al.*, 2018; Fisher and Elbaum-Garfinkle, 2020; Greig *et al.*, 2020; Martin *et al.*, 2020; Bremer *et al.*, 2022). In addition, it has been demonstrated that some residues act as 'stickers' and promote phase separation, while other residues known as 'spacers' favour the solubility of proteins (Harmon *et al.*, 2017, 2018; Holehouse and Pappu, 2018a). While at first glance, some stickers may contain similar functional groups, they can be unequal contributors to biomolecular phase separation. For instance, Y is better than F and R is better than lysine (K) in stabilising condensates (Nott *et al.*, 2015; Brady *et al.*, 2017; Lin *et al.*, 2017; Qamar *et al.*, 2018; Wang *et al.*, 2018; Fisher and Elbaum-Garfinkle, 2020; Greig *et al.*, 2020; Martin *et al.*, 2020; Bremer *et al.*, 2022). R may also modulate phase separation in a context-dependent manner (Bremer *et al.*, 2022). Some of these observations raise an important question: what are the key features that characterise the underlying energy landscapes of phase-separating proteins? In this paper, we address this question by applying the energy landscape framework to peptides with different sequences encoding $\pi$–$\pi$ and cation–$\pi$ interactions that are known to promote phase separation of proteins yielding biomolecular condensates. The energy landscape framework allows us to explore the potential energy landscape of the peptides by performing geometry optimisation to identify local minima and transition states, and connecting them via steepest-descent pathways (Wales, 2003). This approach provides a powerful

tool to explain emergent observable properties in terms of the atomic interactions at a fundamental level.

Specifically, we performed a computational analysis of the potential energy landscape for various hexapeptide monomers modelled at the atomic scale. We chose hexapeptides because the secondary structure of pentapeptides is context-dependent, that is, the same sequence of five amino acids can occur in different secondary structures, such as $\alpha$-helix and $\beta$-sheet (Kabsch and Sander, 1984). Therefore, hexapeptides may represent the minimal system useful for investigating the conformational properties of peptides, as well as the intramolecular interactions between the amino acids within a peptide monomer. In the stickers-and-spacers model, the 'stickers' are the interaction sites that can either be single amino acids, groups of residues, or entire domains that promote phase separation, and 'spacers' favour the solubility of proteins (Harmon *et al.*, 2017; Holehouse and Pappu, 2018b; Yang *et al.*, 2019). Following the stickers-and-spacers model, the hexapeptides are chosen to contain two dipeptide stickers joined together by a glycine–glycine (GG) spacer (Abbas *et al.*, 2021). Working with such minimal systems allows us to directly link the differences in the energy landscapes to specific interactions between amino acid pairs, and hence, reduce the impact of cooperative and competitive effects.

A key signature of the energy landscape of a molecule is its heat capacity ($C_V$). Previous simulations of clusters have shown that low-temperature peaks in $C_V$ represent solid–solid transitions between alternative low-energy conformations that differ significantly in terms of their enthalpy and entropy (Doye and Wales, 1995, 1998; Doye *et al.*, 1998; Doye and Calvo, 2002; Bogdan *et al.*, 2006). In this contribution, we exploit the capability to produce rapid analysis of the heat capacity and assign the peaks to specific local minima with distinct intramolecular interactions. Measurement of $C_V$, as a function of temperature, can be useful to gain better insight into the thermodynamic properties of biopolymers, using differential scanning calorimetry (Benzinger, 1971; Poland, 2001, 2002; Prabhu and Sharp, 2005; Cooper, 2010). In general, low temperature $C_V$ measurement is useful for entropy calculation (Giauque and Johnston, 1929) and for accessing vibrational modes of the molecule that are otherwise inaccessible to spectroscopic techniques that provide information about optical vibrational modes. These modes provide information about molecular conformations and stabilising interactions (Mrevlishvili, 1979). Even though biological molecules are not functional at extreme temperatures, thermodynamic analysis can offer new insights into the properties and behaviour that may have relevance at physiological temperatures. This analysis is similar to the study of crystalline (Starkweather, 1960) and amorphous polymers at low temperatures (Warfield and Petree, 1962). Specific heat measurements for peptides at low temperatures (1.8–20 K) can be employed as a measure of the elasticity of the molecule (Finegold and Cude, 1972). Here, we calculate the $C_V$ of peptide monomers using the harmonic superposition approximation (Wales, 2017), and we observe features (peaks or inflection points) at low temperatures for the hexapeptides with phase separation promoting residues. The low-temperature peaks arise from competing structural motifs for a relatively small number of low-lying local minima. Peaks can be assigned to competition between these minima using the temperature derivative of the occupation probability (Wales, 2017). The theory provides an exact decomposition of $C_V$ in terms of local minima within the same approximation, which reveals the important cases of interest, where the peaks arise from competition between a few low-energy conformations. We emphasise that peaks

in $C_V$ are simply being used as a diagnostic of the structure in the underlying landscape. This structure is clear in the harmonic normal mode approximation to the partition function; a more accurate treatment of $C_V$ is not required to achieve this diagnostic.

The degree of frustration (Bryngelson and Wolynes, 1987; Onuchic and Wolynes, 2004) of the potential energy landscape, quantified via a frustration metric (De Souza *et al.*, 2017), reveals the persistence of high energy barriers separating low-lying minima. In other words, the frustration reflects the existence of competing configurations. The frustration is caused by different low-lying potential energy minima separated by significant barriers. Here, we find that the landscape is more frustrated for the peptides that contain residues (Y/R) with a higher propensity for phase separation, compared to the residues with a lower phase separation propensity. This observation agrees with the finding that the potential energy landscapes for intrinsically disordered proteins are multi-funnelled (Chebaro *et al.*, 2015). However, the frustration metric (De Souza *et al.*, 2017) alone is not sufficient to predict phase separation propensity. Overall, we observe that the peptides with residues that have high phase separation propensity have distinct peaks or inflection points at low temperatures (significantly below the melting temperature) in $C_V$ plots and more frustrated potential energy landscapes. These features in $C_V$ correspond to competing structures stabilised by alternative interactions (aromatic–aromatic or cation–aromatic), or where the residues are oriented differently. This analysis suggests that the calorimetric criterion is a necessary but not a sufficient condition for phase separation (Zhou *et al.*, 1999). The frustration metric provides an additional diagnostic to compare the phase separation propensity of residues in sequences that already exhibit features in $C_V$ at low temperatures.

## Methods

The workflow adopted during the current study is presented in Fig. 1 and summarised below. The peptide sequences are constructed using the stickers-and-spacers model (Holehouse and Pappu, 2018b), and the hexapeptides are modelled using the FF99IDPs (Case *et al.*, 2005; Wang *et al.*, 2014) force field (Step 1, Fig. 1). The FF19SB (Tian *et al.*, 2020) potential was also tested for some of the peptides to ensure that the structures represented by $C_V$ features depend on the interactions within the sequence and not on the force field (Supplementary Material). The potential energy landscape is then explored using basin-hopping parallel tempering (BHPT; Step 2, Fig. 1) (Li and Scheraga, 1987, 1988; Wales and Doye, 1997; Strodel *et al.*, 2010). Discrete path sampling (Wales, 2002) is employed to find the connected pathways between local minima (Step 3, Fig. 1). The convergence of sampling is monitored via disconnectivity graphs (Becker and Karplus, 1997; Wales *et al.*, 1998) and heat capacities. The $C_V$ analysis is performed using the harmonic superposition approximation (Step 4, Fig. 1) (Wales, 2017), and the frustration in the landscape is quantified via a frustration metric (De Souza *et al.*, 2017).

### Peptide model using AMBER

The hexapeptides are modelled using a properly symmetrised (Malolepsza *et al.*, 2010) version of the FF99IDPs (Wang *et al.*, 2014) force field along with an implicit solvent model (igb = 8), and a monovalent ion concentration of 0.1 M (Case *et al.*, 2005, 2022). The N- and C-terminals are methylated and

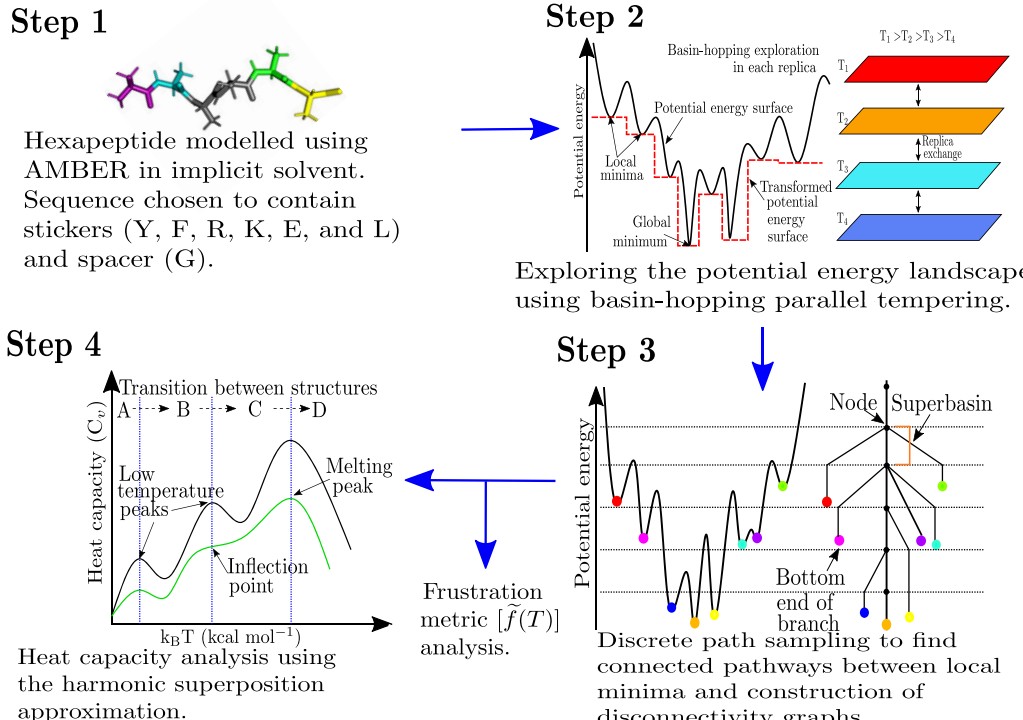

**Figure 1.** Schematic figure representing the workflow for the computational potential energy landscape exploration to interrogate peptides of varying phase separation propensities.

methylamidated, respectively, to cap the charges in the zwitter-ionic form of the peptide (Step 1, Fig. 1). We also tested another force field, FF19SB (Tian *et al.*, 2020), and the uncapped peptides for both the force fields. The corresponding results are presented in the Supplementary Material.

### Basin-hopping parallel tempering

The global optimisation program GMIN (Wales, 2023a) is used to perform basin-hopping (Wales and Doye, 1997; Strodel *et al.*, 2010; Li and Scheraga, 1987, 1988). For the current computation, the AMBER interface with GMIN is employed. A total of 16 replicas are used with temperatures exponentially distributed between 300 and 575 K. The exchanges are attempted at random with a mean frequency of 10, that is, an average of one exchange every 10 steps. The potential energy landscape is explored by performing 100,000 Cartesian coordinate steps and group rotation (Mochizuki *et al.*, 2014) moves for the side chains. The local minima with $C_\alpha$ in D-form and peptide bonds as *cis*-isomer are discarded. A root-mean-square (RMS) force convergence criterion of $10^{-7}$ kcal/(mol Angstrom) is employed to save the 400 lowest energy structures differing by at least 0.01 kcal mol$^{-1}$ (to ensure uniqueness of local minima) after running BHPT (Step 2, Fig. 1) (Strodel *et al.*, 2010).

### Discrete path sampling

Discrete path sampling (Wales, 2002) implemented in the OPTIM (Wales, 2023b) and PATHSAMPLE (Wales, 2023c) programs is used to find optimal pathways between the local minima and the global minimum. A discrete path is defined as an elementary rearrangement between a local minimum, transition state, and another local minimum. The local minimum is defined as a stationary point with no negative Hessian eigenvalues, whereas a

transition state is a first-order saddle point with exactly one negative Hessian eigenvalue (Murrell and Laidler, 1968; Wales, 2003). The doubly-nudged (Trygubenko and Wales, 2004) elastic-band algorithm (Henkelman and Jónsson, 2000; Henkelman *et al.*, 2000) is used to generate candidate transition states, which are then refined accurately using hybrid eigenvector-following (Munro and Wales, 1999). Approximate steepest-descent is employed to find the local minima connected by the transition state using the limited-memory Broyden–Fletcher–Goldfarb–Shanno (L-BFGS) algorithm (Nocedal, 1980; Liu and Nocedal, 1989). Dijkstra's shortest path algorithm (Dijkstra, 1959) is then used to choose the next pair of minima for which a new connection attempt is made, and the process is repeated until a fully connected pathway is found between the minima of interest using the missing connection algorithm (Carr *et al.*, 2005). In the case of some peptides, chain crossing is observed. For these peptides, quasi-continuous interpolation (QCI) (Wales and Carr, 2012; Röder and Wales, 2018) is employed to find the correct pathways. Finally, the stationary point database is optimised using the UNTRAP procedure (Strodel *et al.*, 2007) in PATHSAMPLE to remove artificial frustration in the landscape; that is, low-lying minima separated by large barriers where a lower energy transition state exists. The convergence of the stationary point database is monitored by the convergence of low-temperature peaks in $C_V$ plots, and by analysing the disconnectivity graph (Step 3, Fig. 1).

### Disconnectivity graphs

The potential energy landscape of a system of $N$ atoms lies in a $(3N + 1)$-dimensional space. Disconnectivity graphs provide a powerful way to visualise the multi-dimensional potential energy landscape (Becker and Karplus, 1997; Wales *et al.*, 1998). They

preserve the information about the minimum barrier for transitions between minima. The vertical axis of the disconnectivity graph represents the potential (or free) energy. The nodes on the vertical axis represent superbasins composed of disjoint sets of minima. Minima lying within the same superbasin can interconvert via a barrier less than or equal to the energy represented by the superbasin. Each branch originates from a node representing the superbasin and terminates at the energy of a local minimum corresponding to a single branch (Step 3, Fig. 1).

### Heat capacity analysis

The harmonic superposition approximation (which is accurate at low temperatures) can be used to express the total partition function as a sum of partition functions of all the local minima. The individual partition functions for the local minima are obtained using normal mode analysis, which yields the harmonic approximation to the vibrational density of states. $C_V$ can now be expressed in terms of occupation probabilities of local minima and their temperature derivatives (Wales, 2017), that is,

$$C_V = \kappa k_B + k_B T^2 \sum_\alpha g_\alpha(T) \left( \frac{\partial \ln p_\alpha(T)}{\partial T} \right)$$

$$= \kappa k_B + \overset{g_\alpha(T)<0}{\sum_\alpha} g_\alpha(T)(V_\alpha - \langle V \rangle_{min}) + \overset{g_\alpha(T)>0}{\sum_\alpha} g_\alpha(T)(V_\alpha - \langle V \rangle_{min})$$

$$\equiv \kappa k_B + C^-(T) + C^+(T).$$

Here, $C_V$ is the heat capacity, $\kappa = 3N - 6$ is the number of vibrational degrees of freedom for a system of $N$ atoms, $k_B$ is the Boltzmann constant, $g_\alpha(T) \equiv \partial p_\alpha(T)/\partial T$ is the derivative of the occupation probability $p_\alpha$ for minimum $\alpha$ with respect to temperature $T$, $V_\alpha$ is the potential energy of minimum $\alpha$, and $\langle V \rangle_{min}$ is the mean potential energy of the minima. The peaks in $C_V$ represent transitions between states with decreasing $(g_\alpha(T)<0)$ and increasing $(g_\alpha(T)>0)$ occupation probability (Wales, 2017).

### Frustration metric calculation

Competing low-energy minima separated by significant barriers make the potential energy landscape frustrated. The frustration of the potential energy landscape can be quantified using a frustration metric ($\tilde{f}(T)$), which is a function of temperature:

$$\tilde{f}(T) = \sum_{\alpha \neq gmin} \frac{p_\alpha^{eq}(T)}{1 - p_{gmin}^{eq}(T)} \left( \frac{V_\alpha^\dagger - V_{gmin}}{V_\alpha - V_{gmin}} \right).$$

Here, $\tilde{f}(T)$ is the frustration metric at temperature $T$, $V_{gmin}$ is the potential energy of the global minimum in the database, $V_\alpha$ is the potential energy of minimum $\alpha$, $V_\alpha^\dagger$ is the potential energy of the highest energy transition state on the lowest energy pathway between $\alpha$ and the global minimum, and $p_\alpha^{eq}$ and $p_{gmin}^{eq}$ are the equilibrium occupation probabilities of minimum $\alpha$ and the global minimum, which are calculated using the harmonic vibrational density of states. The global minimum does not contribute to frustration and its inclusion leads to an erroneous decrease in frustration at low temperature. Hence, the global minimum is excluded from the frustration metric calculation and occupation probabilities of the remaining minima are renormalised (De Souza *et al.*, 2017).

### Results and discussion

The importance of multivalency (Li *et al.*, 2012), interaction strength (Asherie *et al.*, 1996; Das and Pappu, 2013; Hyman *et al.*, 2014; Brangwynne *et al.*, 2015; Choi *et al.*, 2020), and accessibility (Ruff *et al.*, 2022) of stickers in promoting phase separation is well established. Here, we explore the energy landscapes (Fig. 2) of various hexapeptides containing a pair of

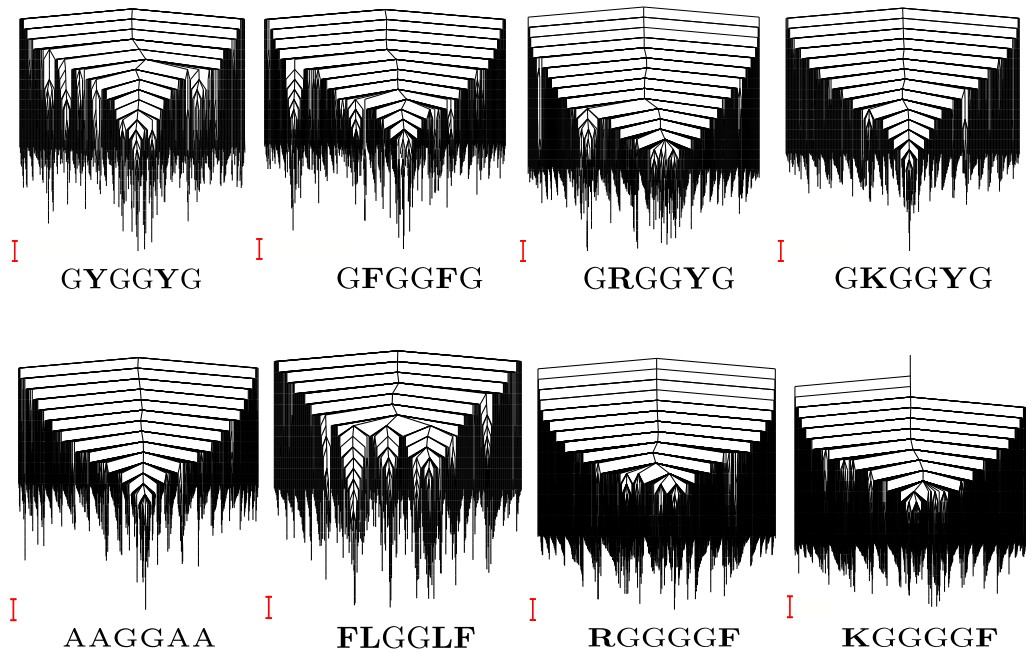

**Figure 2.** Representative disconnectivity graphs (Becker and Karplus, 1997; Wales *et al.*, 1998) for some of the peptides studied. The scale bar is 1 kcal mol$^{-1}$.

dipeptide stickers separated by a GG spacer. The dipeptide stickers include FF, YY, RY, KY, YR, YK, RE, KE, FL, LF, and LL (Abbas *et al.*, 2021). These sequences are chosen to encode the aromatic–aromatic, cation–aromatic, cation–anion and CH–$\pi$ interactions. The interactions between individual pairs of amino acids are further interrogated by analysing hexapeptides with a pair of stickers separated by two or four glycines. Energy landscapes are also explored for poly-amino acid hexapeptides containing a single type of amino acid residue, including alanine (A), glycine (G), valine (V), arginine (R) and lysine (K). The peptides containing residues with better phase separation propensity show clear features in $C_V$ at low temperatures (Fig. 3a, see section "Heat capacity at low temperature"). These features are caused by competing low-energy conformations with different types of interactions (Figs 4 and 5, see section "Interactions leading to features in $C_V$"). Further analysis of frustration reveals that the peptides with amino acids encoding better phase separation propensity result in more frustrated landscapes (Fig. 3b, see section "Frustration in the energy landscape"). It is hypothesised that the collective behaviour of phase separation may be understood in terms of single-molecule properties by quantifying the heat capacity and frustration within the energy landscape

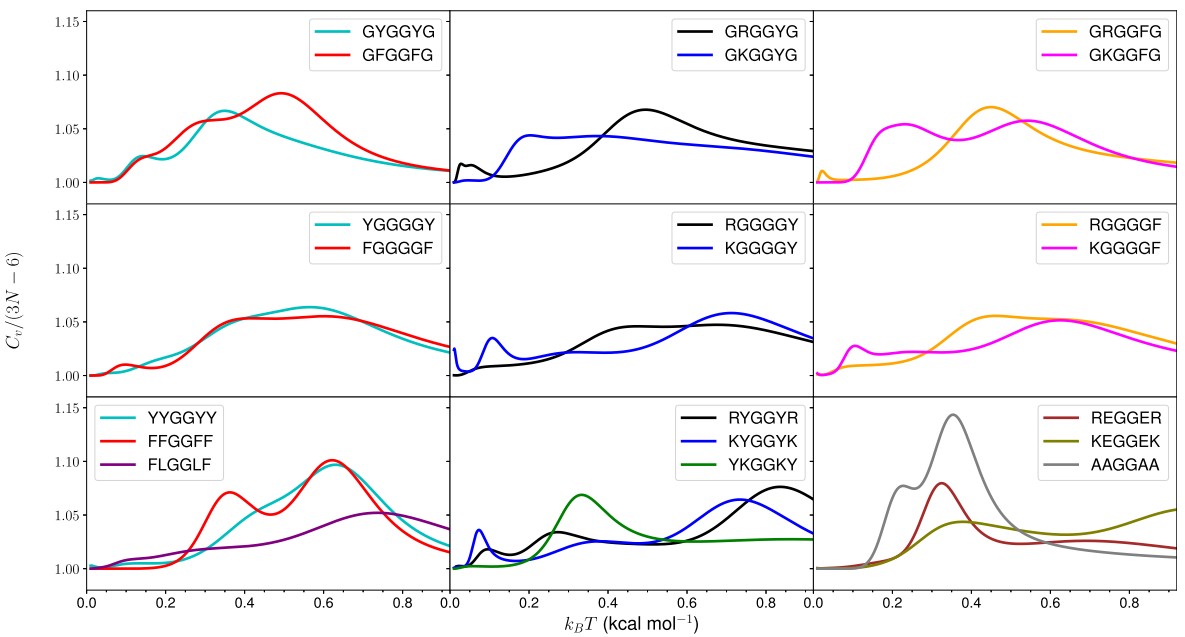

(a) Heat capacity in kcal/(mol K) versus $k_B T$ in kcal mol$^{-1}$ for various hexapeptides.

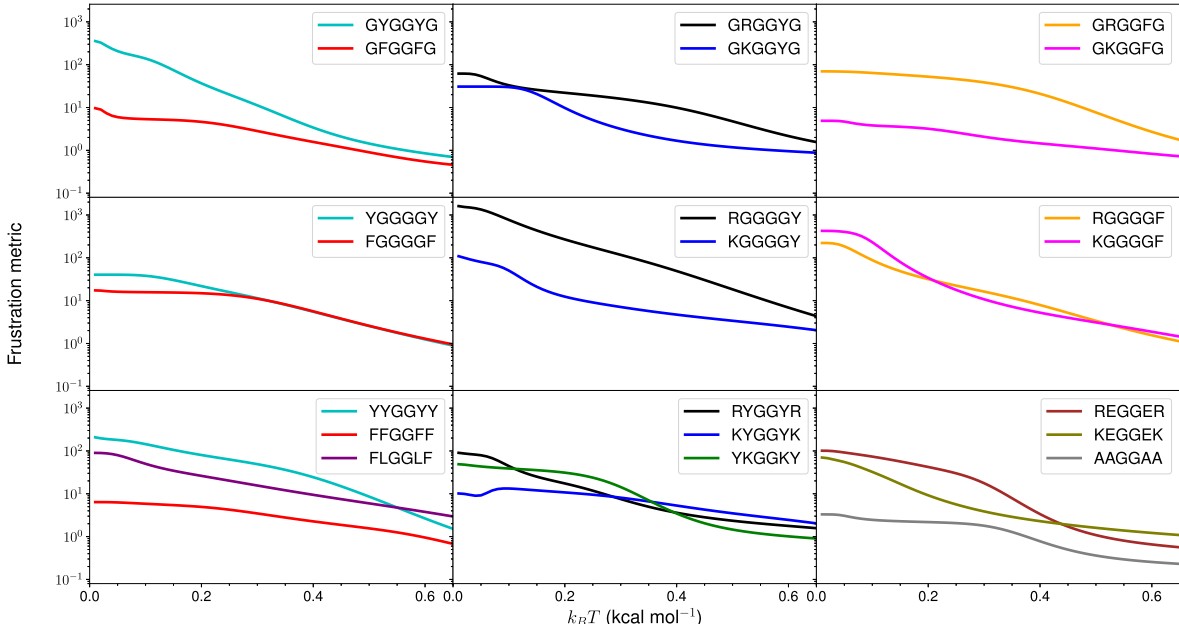

(b) Frustration metric $[\widetilde{f}(T)]$ (De Souza et al., 2017) versus $k_B T$ in kcal mol$^{-1}$ for various hexapeptides.

**Figure 3.** Heat capacity and frustration metric diagnostic for probing phase separation propensity encoded by different amino acid residues.

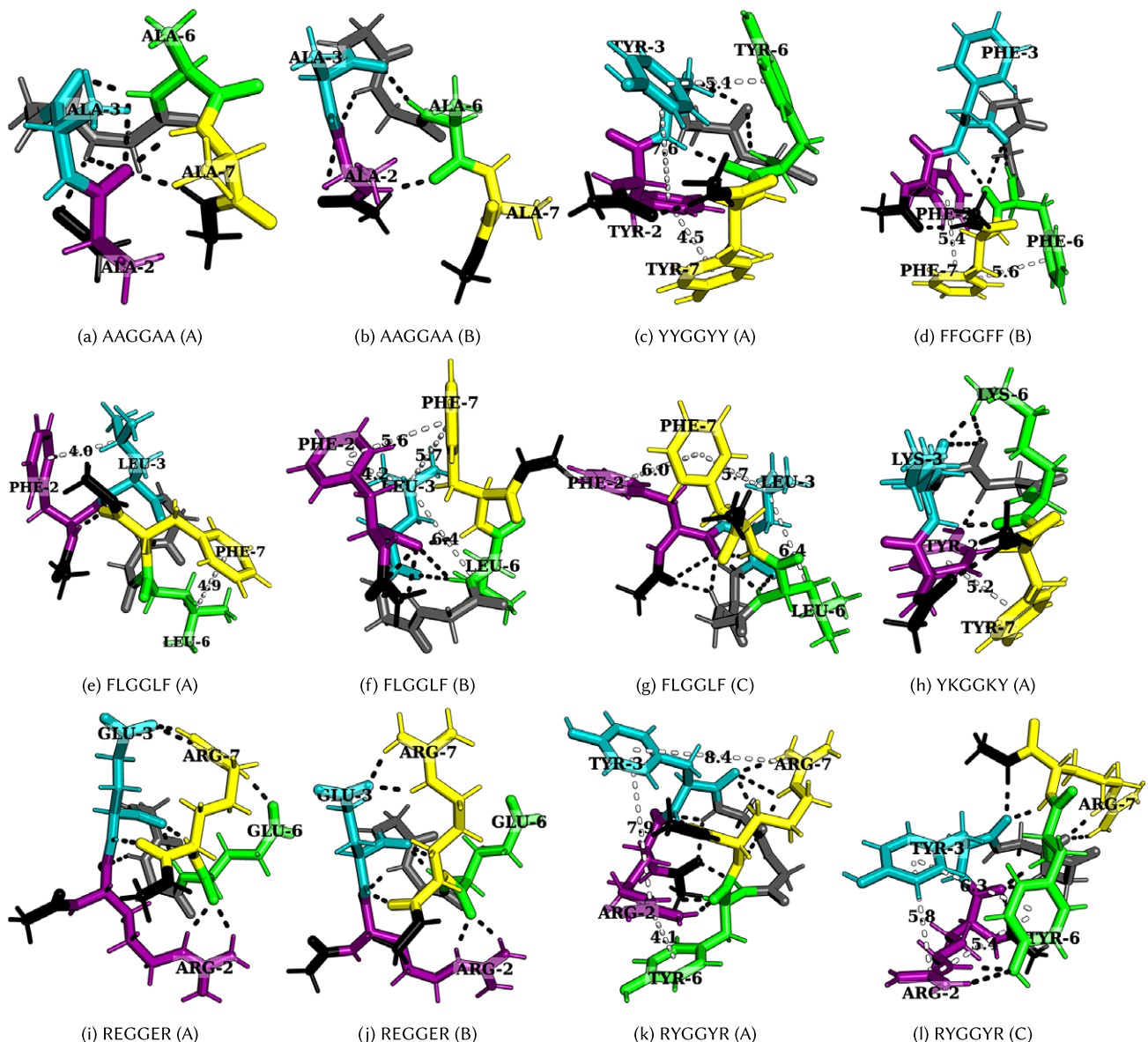

**Figure 4.** Structures corresponding to low-temperature heat capacity features. The first and second peaks correspond to the transition from A to B and then from B to C, respectively.

framework. An interesting analogue is how the existence of different conformations leads to polymorphic forms for various organic and inorganic molecules (Supplementary Material). A recent study has also shown links between heat capacity change during unfolding and multicomponent phase separation behaviour (Rana *et al.*, 2023).

### *Heat capacity at low temperature*

We first investigate the geometric and energetic parameters that underlie the structural differences represented by low-temperature peaks in the heat capacity of peptides with varying phase separation propensities (Fig. 3*a*). We emphasise that we are using these features as a diagnostic for competing structures in the energy landscape, which may correlate with phase separation propensity. This computational construction does not need to

be an accurate calculation of $C_V$, nor does it need to be experimentally accessible. These peaks represent the transition between competing structures that have significant enthalpic and entropic differences and the integral over the peak represents the latent heat for this transition. In some $C_V$ plots, instead of distinct peaks, we observe inflection points (GFGGFG, YGGGGY, RGGGGY, RGGGGF, YYGGYY, and FLGGLF) where the curvature of the plot changes. These inflection points (or shoulders) may be caused by overlapping peaks. The temperatures corresponding to these distinct inflection points are also considered, since they may contain useful information. The hexapeptide AAGGAA is taken as the control, as it is predicted to have the lowest phase separation propensity of the set (Wang *et al.*, 2018), and the corresponding $C_V$ is simpler (the potential energy landscape is not frustrated) compared to other peptides with more phase separation promoting residues. Note that it is not the height of the peaks but the

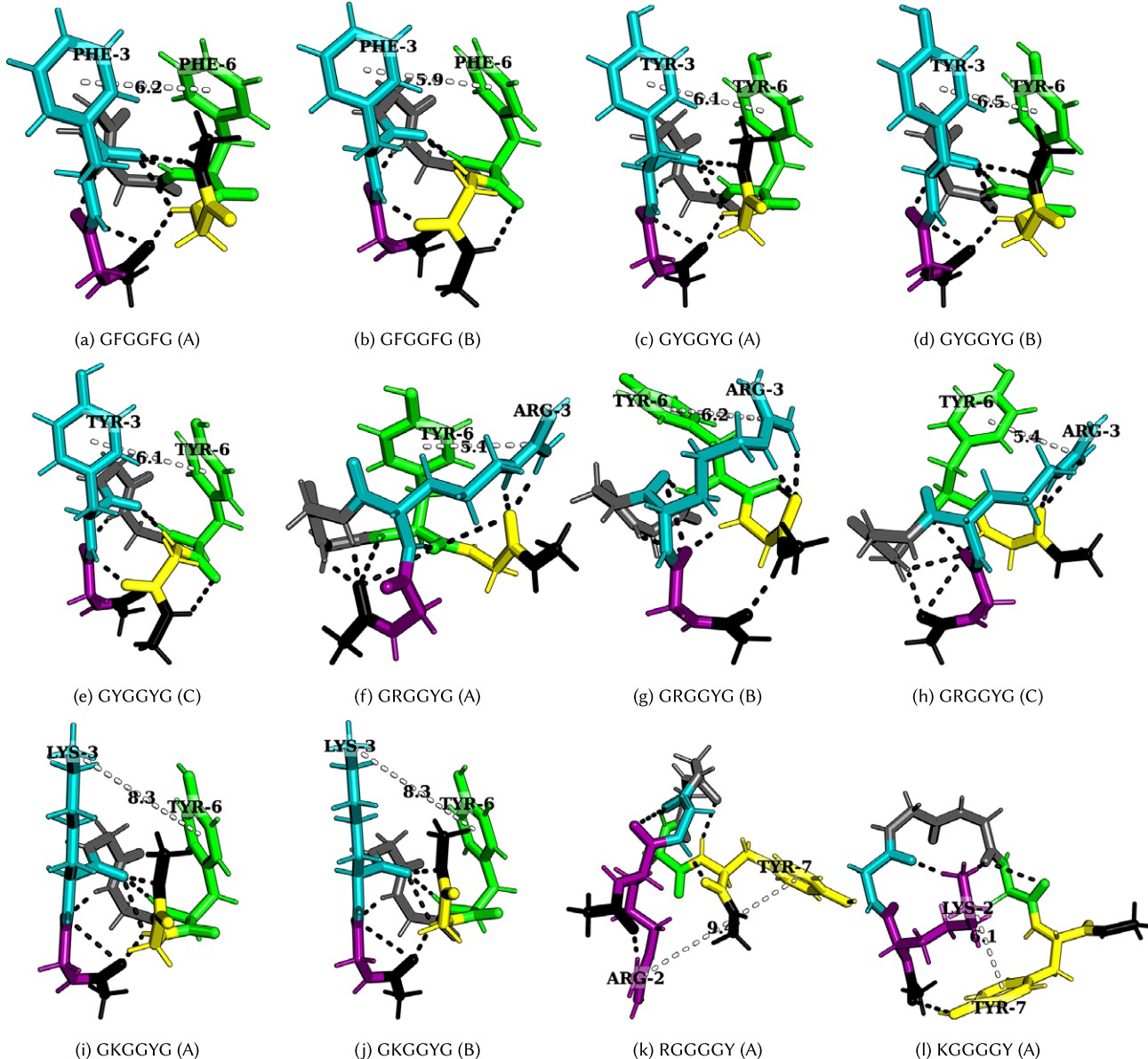

**Figure 5.** Structures corresponding to low-temperature heat capacity features. The first and second peaks correspond to the transition from A to B and then from B to C, respectively.

existence of features at low temperature (below the melting temperature) that report on the structural heterogeneities in the landscapes, and hence, the phase separation propensities of the constituent residues in a sequence.

Various other hexapeptides, such as GGGGGG, AAAAAA, VVVVVV, EEEEEE, RRRRRR, and KKKKKK, have also been analysed as controls and are found to show simpler $C_V$ profiles (Supplementary Material). However, distinct polar contacts between the main-chain atoms or between the main-chain and side-chain atoms can produce features in $C_V$ (AAGGAA in Fig. 4a,b).

In general, for hexapeptides with interactions that encode a higher propensity for phase separation, we observe more pronounced features (several distinct peaks and inflection points) in $C_V$. The frustration metric can then be used as a further diagnostic. The $C_V$ plots for various other peptides are given in the Supplementary Material.

### Interactions leading to features in $C_V$

A low-temperature heat capacity peak often arises from a transition from a compact structure with two sets of dominant interactions between four residues (YYGGYY – Fig. 4c and FLGGLF – Fig. 4e) to another structure with a similar set of interactions, but with residues oriented differently, or a relatively extended structure with two sets of dominant interactions between three residues (FFGGFF – Fig. 4d and FLGGLF – Fig. 4f,g). Depending on the number of stickers in the peptide, the low-temperature peak may also correspond to a transition from two sets of dominant interactions between three residues to a single principal interaction between two residues (KEGGEK and REGGER – Fig. 4i,j). A detailed discussion of the competing structures for various hexapeptides is given below.

*Tyrosine versus phenylalanine:* The presence of a hydroxyl group in tyrosine not only enhances its hydrogen-bonding ability, but also

results in different rotamers, leading to features in $C_V$ at low temperatures (GYGGYG and YGGGGY). GFGGFG and GYGGYG exhibit inflection points and distinct peaks at low temperatures, respectively. In particular, the low-temperature feature in GFGGFG and FGGGGF corresponds to the transition between a structure with methyl–aromatic and aromatic–aromatic interactions to a structure with an aromatic–aromatic interaction, which further changes to a structure with several polar contacts between distinct atoms. In contrast, for GYGGYG and YGGGGY, the features at low temperature correspond to the transition between rotamers of the aromatic ring containing methyl–aromatic and aromatic–aromatic interactions. Here, the methyl group belongs to the C-terminal cap of the peptide. Interestingly, the observation of a low-temperature peak resulting from the presence of ring rotamers can be compared to an experimental observation in which a bulge in the $C_V$ plot of polystyrene was attributed to the rotation of the phenyl ring around the chain axis (Warfield and Petree, 1962). The orientation that optimises the aromatic interaction depends on the distance between the $C_\alpha$ atoms, stacked at a short distance and T-shaped at a longer distance (Hunter et al., 1991; Chelli et al., 2002). Offset-stacked structures can also be energetically favourable (Ninković et al., 2014), and the methyl group of the cap can also interact with an aromatic residue (Zanuy et al., 2004). We observe similar edge-to-face, CH–$\pi$, and methyl–aromatic interactions for GFGGFG (Fig. 5a,b), GYGGYG (Fig. 5c–e), FGGGGF, and YGGGGY.

*Arginine versus lysine*: GRGGYG exhibits features in $C_V$ because of the interaction between R and Y, and the presence of ring rotamers (rotamer of an aromatic ring) for Y (Fig. 5f–h), whereas for GKGGYG and GKGGFG, it is the methyl group in the C-terminal cap that preferably interacts with the Y/F (Fig. 5i,j). We still see features in $C_V$ for GKGGYG because of ring rotamers for Y. In the case of RYGGYR, one of the peaks corresponds to the structural transition between the aromatic–cation–aromatic interaction motif to the aromatic–cation interaction motif (Fig. 4k,l). Hence, R has more propensity than K to interact with the aromatic residues.

*Context-dependence*: Phase separation may be regarded as a percolation network transition (Mittag and Pappu, 2022). In other words, the formation of a stable condensate occurs when biomolecules interconnect with one another forming a percolated network; the denser the connectivity of the percolated network, the higher the stability of the condensates (Espinosa et al., 2020). The difference in size, the steric packing of R and K, the number of spacers between the stickers, and the distance between the stickers may be useful in explaining the context-dependent properties of these amino acid residues in terms of accessibility and networking ability of stickers to interact with each other. Consider the peptides RYGGYR, GRGGYG, GKGGYG, RGGGGY, and KGGGGY. Even though the presence of R leads to more features in the $C_V$ plots, we observe that when the cationic and aromatic residues are far apart, as for RGGGGY and KGGGGY, K seems to be more flexible and less sterically inhibited, and therefore, it can interact well with Y, whereas R seems to be more rigid and does not interact favourably with Y/F (Fig. 5k,l). Previous reports suggest that K/RNA coacervates are more dynamic than R/RNA coacervates (Ukmar-Godec et al., 2019), and the R-rich motif may act as a phase disruptor (Odeh and Shorter, 2020). While the different behaviours of R and K may be understood in terms of the relative strength of the interactions, it is also possible that the flexible nature of K compared to R may play a role. Furthermore, the shuffling of sequence may alter the presence of charged residues near the N-/C-termini, which may lead to differences in the properties of these peptides because of the charge interaction with the peptide dipole. The dipole moment

effect is expected to be more significant in the case of an uncapped peptide in zwitterionic form (Marqusee and Baldwin, 1987; Tkatchenko et al., 2011).

*Aromatic–aromatic versus cation–aromatic interactions*: Favourable cation–aromatic interactions between R and Y are observed in RYGGYR (Fig. 4k,l). However for YKGGKY, the aromatic–aromatic interaction between two tyrosine residues is preferred over the cation–aromatic interaction between K and Y (Fig. 4h). This observation hints at the role played by the proximity of interacting residues in a sequence, that is, the two tyrosine residues located at the ends can establish an aromatic–aromatic interaction, which is preferred over the weaker interaction offered by the lysine residues. From a broader perspective, this result may be useful in understanding the context-dependent properties of amino acid residues across different sequences.

*Cation–anion interaction*: Hydrogen-bonding between oppositely charged amino acids may lead to the formation of salt bridges where the same residue interacts with two different residues (complex) or between two oppositely charged residues (simple) (Musafia et al., 1995). Both REGGER (Fig. 4i,j) and KEGGEK exhibit low-temperature $C_V$ peaks corresponding to the transition from structures containing a complex salt bridge to a simple salt bridge. The complex salt bridge is formed by the interaction of the same cationic residue with two anionic residues. The next $C_V$ peak at a higher temperature corresponds to the transition from a structure with a cation interacting with a particular anion to a structure with the same cation interacting with a different anion in a different orientation, as in the case of uncapped KEGGEK peptide.

*Partial phase separation*: Leucine and phenylalanine are constituents of peptides exhibiting partial phase separation (Abbas et al., 2021), and the $C_V$ plot for FLGGLF contains features at low temperatures. The peak represents the transition from a structure containing two distinct pairs of L–F interactions, arising from four residues, to a structure with two pairs of interactions arising from three residues F, F, and L (Fig. 4e–g). Several CH–$\pi$ interactions can occur between L and F. Hence, partial phase separation may occur for peptides containing amino acids capable of exhibiting distinct pairs of interactions. However, the interaction strength between stickers is weaker compared to the cation/aromatic–aromatic interaction. Although weak, the CH–$\pi$ interaction is known to play an important role in supramolecular organisation (Piccolo, 2001).

### Frustration in the energy landscape

The frustration (Bryngelson and Wolynes, 1987; Onuchic and Wolynes, 2004) in the multi-dimensional potential energy landscape can be visualised by analysing the multiple funnels in the disconnectivity graph representation (Becker and Karplus, 1997; Wales et al., 1998) (Fig. 2). More funnels with low-energy minima separated by significant barriers from the global minimum make the landscape more frustrated at low temperatures. The frustration is high at very low temperatures because the molecules do not have enough thermal energy to overcome the barrier required for transition from one low-energy conformation to another. In other words, if the state of system corresponds to a low-energy minimum in one funnel, the system is likely to remain in the same funnel when the frustration is high. At higher temperatures the thermal energy is larger and so the molecules have sufficient energy to overcome the barriers and transition between local minima. Hence, the system is less frustrated at higher temperatures. Quantitatively, the frustration metric (De Souza et al., 2017) is generally larger for peptides containing Y/R than for peptides containing F/K at lower temperatures (Fig. 3b). In particular, at a very low temperature corresponding to $k_B T =$ 0.2 kcal mol$^{-1}$, the

frustration metric for GYGGYG is 8 times the value for GFGGFG, YYGGYY is 16 times larger than FFGGFF, GRGGYG is 2 times larger than GKGGYG, GRGGFG is 16 times larger than GKGGFG, and REGGER is 5 times larger than KEGGEK. At $k_B T = 0.1$ kcal mol$^{-1}$ the frustration metric of RYGGYR is 3 times greater than that of KYGGYK. Hence, it appears that the frustration in the landscape for the monomer peptide directly correlates with the relative phase separation propensity of its constituent residues. This result can also be rationalised by correlating the high frustration with the tendency to be trapped in the unfolded state, and it is well known that unfolded states and intrinsically disordered proteins promote phase separation (Majumdar *et al.*, 2019). Interestingly, KGGGGF is three times more frustrated than RGGGGF at very low temperature ($k_B T = 0.1$ kcal mol$^{-1}$). Here, the larger number of spacers (four glycines) increases the distance between the stickers and affects the inaccessibility. The accessibility is reduced more in the case of R, which appears more rigid compared to the more flexible K residue. This difference may explain the context-dependent properties of R in phase-separating proteins. Moreover, the potential energy landscape of FLGGLF is five times more frustrated than FFGGFF at a very low temperature ($k_B T = 0.2$ kcal mol$^{-1}$). However, FFGGFF has distinct peaks in the $C_V$, in contrast to FLGGLF (Fig. 3*a*). These features are caused by a stronger aromatic–aromatic interaction between two F, which correlates with the better phase separation propensity of residues in FFGGFF, whereas the interaction between F and L may facilitate partial phase separation (Abbas *et al.*, 2021). The frustration metric plots for various other peptides are given in the Supplementary Material.

## Conclusions

We have investigated the hypothesis that the energy landscape of peptide monomers may report on their phase separation ability, which is a collective phenomenon. The different possible arrangements in which the aromatic–aromatic and cation–aromatic interactions can occur in a peptide monomer can produce low-temperature peaks in the heat capacity. Additionally, the high barriers between the alternative low-lying potential energy minima and the existence of several such conformations, as visualised by multiple funnels in the disconnectivity graph, produce a highly frustrated potential energy landscape. Together, features in the heat capacity plot, and the frustration of the landscape, quantified using the frustration metric, appear to correlate with increased phase separation propensity of the constituent residues. The high frustration results from the molecule being trapped in an intrinsically disordered or unfolded state, and both these states are known to promote phase separation.

This analysis also provides a useful framework to investigate the context-dependent properties of amino acid residues in different sequences. While there have been several attempts (Dzuricky *et al.*, 2020; Simon *et al.*, 2017) to guide the rational design of peptides useful for bioengineering applications, the present study presents a new perspective to design peptides with targeted phase separation behaviour. A related study provides links between the secondary structures that contribute to low-temperature $C_V$ features for monomers and dimers of hexapeptide sequences that are experimentally known to aggregate (Nicy and Wales, 2023). It is important to understand that we are not actually interested in the low-temperature behaviour of the heat capacity and that an accurate calculation is not required. Rather, we are using peaks in an approximate $C_V$ as a computational construction to diagnose competition between alternative favourable structures. It is the characteristics of these conformations that may provide a structural interpretation and diagnostic of higher-order behaviour in condensates, such as liquid–liquid phase separation. Our results suggest that there may indeed be such a connection. We do not claim that this connection is universal, but we do suggest that it may be useful.

**Open peer review.** To view the open peer review materials for this article, please visit http://doi.org/10.1017/qrd.2023.5.

**Supplementary material.** The supplementary material for this article can be found at https://doi.org/10.1017/qrd.2023.5.

**Data availability statement.** The discrete path sampling databases are available at https://doi.org/10.17863/CAM.96972 (Nicy *et al.*, 2023). The step-by-step protocol for creating one such database is given as a tutorial on https://github.com/nicy-nicy/peptide-energy-landscape-exploration. The scripts to analyse the databases can be found at https://github.com/nicy-nicy/energy-landscape-cv-analysis.

**Author contribution.** J.A.J., R.C.G., D.J.W., and Nicy conceived the idea and designed the study. Nicy performed the simulations and wrote the first draft. All the authors helped with the analysis, interpretation of data and corrected the final draft.

**Financial support.** This work was supported by Engineering and Physical Sciences Research Council (EPSRC) (D.J.W., Grant No. EP/N035003/1), the Cambridge Commonwealth, European and International Trust, the Allen, Meek and Read Fund, the Santander fund, St Edmund's College, University of Cambridge, and the Trinity-Henry Barlow Honorary Award (Nicy).

**Competing interest.** The authors declare no competing interest exists.

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
