## [Reviewer Report]

*Comments to Author*: I found the physics based approach taken in this paper - e.g. probing the importance of the free energy landscape of short peptides on their phase behaviour - an extremely interesting one. The assumptions (e.g. approximations in the calculation Cv) are robustly discussed and justified in terms of the physics being learned from the study. I found Fig 1 helpful in understanding the simulation methodology. Given that this method would be of interest to a broad range of disciplines, and the importance of membraneless organelles, I would invite the authors to add a few more sentences in specific areas providing readers with a bit more background, so that researchers who are not experts in biological phase separation can enjoy the work.

For example, it would be helpful to briefly explain (e.g. 1/2 sentences) the ideas for why phase separation and intrinsic disorder are related in the introduction.

Please could “the stickers and spacers” model be briefly explained (line 61).

Please could the percolation network model of phase separation be explained (line 269).

I found the Temperature units in Fig 3 rather confusing - would plain Kelvin be clearer to readers? I also wondered why the frustration decreases with temperature, and if the authors might provide a bit more explanation about how this metric is expected to behave. For example, I would have thought that the frustration is zero at 0K when all systems are in their absolute ground state but that doesn’t seem to becorrect. Maybe a couple of toy examples and their high/low temperature limits could be described in the text to help provide the readers with a physical picture of the meaning of the frustration.

The authors are rather cautious and they do not provide any hypothesis as to the thermodynamic mechanisms for why a frustrated free energy landscape would make peptides more likely to phase separate. I would welcome such an addition in the final paragraph, even if the ideas are speculative, or if there are potentially conflicting viewpoints that need to be discussed. Equally, I am aware that this is not considered appropriate in certain areas of soft matter, so I leave it to the authors to judge.

---

## [Reviewer Report]

*Comments to Author*: Wales and co-workers used the simulation techniques developed by the Wales group to characterize the energy landscape of a number of hexapeptides with different phase separation propensities. They first performed global optimization to identify the local minima (including the global one) of each peptide and then connected the different minima using discrete path sampling. The resulting database of interconnected minimum-transition state-minimum collections was visualized using disconnectivity graphs showing the potential energy landscapes. The energy landscape was further characterized in terms of heat capacity and frustration. The main finding is that peptides with phase-separation promoting interactions have energy landscapes that are more frustrated at low temperatures.

This is a fresh view of the problem of protein phase separation and surely worth publishing. The work was conducted at a very high level, and the manuscript is well written.

Nonetheless, I suggest that the authors address the following points, which I believe would further enhance the impact of their work:

1) The discussion of the results is very detailed, going into the individual types of interactions. There is nothing wrong with this; however, what is missing is a summary figure (quantitatively) correlating the determined frustration of the energy landscape with the phase separation propensity of the discussed peptides or interaction types. This would be a good summary figure to keep in mind for a while after reading the paper, and would also make the paper more accessible to readers interested in phase separation but not familiar with the not so common simulation methods of the Wales group.

2) It is well known that unfolded proteins prefer phase separation (e.g., DOI:10.1021/acs.jpclett.9b01731). I wonder, therefore, if the frustration in energy landscapes revealed by the authors correlates with the peptides' aversion to unfold. It would be good if the authors could provide more macroscopic descriptions of the peptide structures, such as their expansion and amount of secondary structure, which could be correlated more generally than the very detailed discussion of the individual peptides, as has been done, with the frustration of the energy landscape and the propensity for phase separation (see my comment 1).

3) The authors mainly focused their discussion of the frustration of the energy landscape at RT=0.2 kcal/mol, which corresponds to a temperature of 100 K. I do not understand how the peptides' behavior at such a low temperature should be decisive for the phase separation at room or physiological temperature. Please elaborate.

4) ll. 56/57: “We chose hexapeptides because the secondary structure of pentapeptides is context-dependent...” The reasoning is difficult to be understood. I guess that the authors want to say that peptides from Nres=5 are able to adopt different secondary structures depending on the environment, which makes sense considering that about 4 residues are required for a complete turn and that the terminal residues are mostly not involved in secondary structure formation.Please rephrase.

5) l. 73: The abbreviation HSA is not used thereafter and could thus be deleted.

6) l. 168 (formula) vs. l. 170: While the choice of the index variable is of course free, it would be nonetheless nice to always use gamma or alpha. I would recommend to switch to alpha in the formula as alpha is also used in the formula further below.

7) l. 192: Is the interaction between pairs (please note the plural) of amino acids or between two amino acids meant?

8) 195: The pepide -> The peptides

9) Fig. 3: At some of the places where the subfigures are joined, the axes labels overlap. Please correct.

---

## [Reviewer Report]

*Comments to Author*: Reviewer #1: I found the physics based approach taken in this paper - e.g. probing the importance of the free energy landscape of short peptides on their phase behaviour - an extremely interesting one. The assumptions (e.g. approximations in the calculation Cv) are robustly discussed and justified in terms of the physics being learned from the study. I found Fig 1 helpful in understanding the simulation methodology. Given that this method would be of interest to a broad range of disciplines, and the importance of membraneless organelles, I would invite the authors to add a few more sentences in specific areas providing readers with a bit more background, so that researchers who are not experts in biological phase separation can enjoy the work.

For example, it would be helpful to briefly explain (e.g. 1/2 sentences) the ideas for why phase separation and intrinsic disorder are related in the introduction.

Please could “the stickers and spacers” model be briefly explained (line 61).

Please could the percolation network model of phase separation be explained (line 269).

I found the Temperature units in Fig 3 rather confusing - would plain Kelvin be clearer to readers? I also wondered why the frustration decreases with temperature, and if the authors might provide a bit more explanation about how this metric is expected to behave. For example, I would have thought that the frustration is zero at 0K when all systems are in their absolute ground state but that doesn’t seem to becorrect. Maybe a couple of toy examples and their high/low temperature limits could be described in the text to help provide the readers with a physical picture of the meaning of the frustration.

The authors are rather cautious and they do not provide any hypothesis as to the thermodynamic mechanisms for why a frustrated free energy landscape would make peptides more likely to phase separate. I would welcome such an addition in the final paragraph, even if the ideas are speculative, or if there are potentially conflicting viewpoints that need to be discussed. Equally, I am aware that this is not considered appropriate in certain areas of soft matter, so I leave it to the authors to judge.

Reviewer #2: Wales and co-workers used the simulation techniques developed by the Wales group to characterize the energy landscape of a number of hexapeptides with different phase separation propensities. They first performed global optimization to identify the local minima (including the global one) of each peptide and then connected the different minima using discrete path sampling. The resulting database of interconnected minimum-transition state-minimum collections was visualized using disconnectivity graphs showing the potential energy landscapes. The energy landscape was further characterized in terms of heat capacity and frustration. The main finding is that peptides with phase-separation promoting interactions have energy landscapes that are more frustrated at low temperatures.

This is a fresh view of the problem of protein phase separation and surely worth publishing. The work was conducted at a very high level, and the manuscript is well written.

Nonetheless, I suggest that the authors address the following points, which I believe would further enhance the impact of their work:

1) The discussion of the results is very detailed, going into the individual types of interactions. There is nothing wrong with this; however, what is missing is a summary figure (quantitatively) correlating the determined frustration of the energy landscape with the phase separation propensity of the discussed peptides or interaction types. This would be a good summary figure to keep in mind for a while after reading the paper, and would also make the paper more accessible to readers interested in phase separation but not familiar with the not so common simulation methods of the Wales group.

2) It is well known that unfolded proteins prefer phase separation (e.g., DOI:10.1021/acs.jpclett.9b01731). I wonder, therefore, if the frustration in energy landscapes revealed by the authors correlates with the peptides' aversion to unfold. It would be good if the authors could provide more macroscopic descriptions of the peptide structures, such as their expansion and amount of secondary structure, which could be correlated more generally than the very detailed discussion of the individual peptides, as has been done, with the frustration of the energy landscape and the propensity for phase separation (see my comment 1).

3) The authors mainly focused their discussion of the frustration of the energy landscape at RT=0.2 kcal/mol, which corresponds to a temperature of 100 K. I do not understand how the peptides' behavior at such a low temperature should be decisive for the phase separation at room or physiological temperature. Please elaborate.

4) ll. 56/57: “We chose hexapeptides because the secondary structure of pentapeptides is context-dependent...” The reasoning is difficult to be understood. I guess that the authors want to say that peptides from Nres=5 are able to adopt different secondary structures depending on the environment, which makes sense considering that about 4 residues are required for a complete turn and that the terminal residues are mostly not involved in secondary structure formation.Please rephrase.

5) l. 73: The abbreviation HSA is not used thereafter and could thus be deleted.

6) l. 168 (formula) vs. l. 170: While the choice of the index variable is of course free, it would be nonetheless nice to always use gamma or alpha. I would recommend to switch to alpha in the formula as alpha is also used in the formula further below.

7) l. 192: Is the interaction between pairs (please note the plural) of amino acids or between two amino acids meant?

8) 195: The pepide -> The peptides

9) Fig. 3: At some of the places where the subfigures are joined, the axes labels overlap. Please correct.